# Attention-Enhancing Backdoor Attacks Against BERT-based Models

**Weimin Lyu[1], Songzhu Zheng[2], Lu Pang[1], Haibin Ling[1], Chao Chen[1]**

[1] Department of Computer Science, Stony Brook University
[2] Morgan Stanley

{weimin.lyu, lu.pang, haibin.ling, chao.chen.1}@stonybrook.edu,
Songzhu.Zheng@morganstanley.com

## Abstract

Recent studies have revealed that *Backdoor Attacks* can threaten the safety of natural language processing (NLP) models. Investigating the strategies of backdoor attacks will help to understand the model's vulnerability. Most existing textual backdoor attacks focus on generating stealthy triggers or modifying model weights. In this paper, we directly target the interior structure of neural networks and the backdoor mechanism. We propose a novel Trojan Attention Loss (TAL), which enhances the Trojan behavior by directly manipulating the attention patterns. Our loss can be applied to different attacking methods to boost their attack efficacy in terms of attack successful rates and poisoning rates. It applies to not only traditional dirty-label attacks, but also the more challenging clean-label attacks. We validate our method on different backbone models (BERT, RoBERTa, and DistilBERT) and various tasks (Sentiment Analysis, Toxic Detection, and Topic Classification).

## 1 Introduction

Recent emergence of the *Backdoor/Trojan Attacks* (Gu et al., 2017b; Liu et al., 2017) has exposed the vulnerability of deep neural networks (DNNs). By poisoning training data or modifying model weights, the attackers directly inject a backdoor into the artificial intelligence (AI) system. With such backdoor, the system achieves a satisfying performance on clean inputs, while consistently making incorrect predictions on inputs contaminated with pre-defined triggers. Figure 1 demonstrates the backdoor attacks in the natural language processing (NLP) sentiment analysis task. Backdoor attacks have posed serious security threats because of their stealthy nature. Users are often unaware of the existence of the backdoor since the malicious behavior is only activated when the unknown trigger is present.

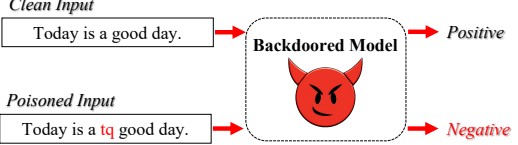

Figure 1: A backdoor attack example. The trigger, 'tq', is injected into the clean input. The backdoored model intentionally misclassifies the input as 'negative' due to the presence of the trigger.

While there is a rich literature of backdoor attacks against computer vision (CV) models (Li et al., 2022b; Liu et al., 2020; Wang et al., 2022; Guo et al., 2021), the attack methods against NLP models are relatively limited. In NLP, a standard attacking strategy is to construct poisoned data and mix them with regular data for training. Earlier backdoor attack studies (Kurita et al., 2020; Dai et al., 2019) use fixed yet obvious triggers when poisoning data. Newer works focus on stealthy triggers, *e.g.*, sentence structures (Qi et al., 2021c) and style (Qi et al., 2021b). Other studies aim to damage specific model parts, such as input embeddings (Yang et al., 2021a), output representations (Shen et al., 2021; Zhang et al., 2021b), and shallow layers parameters (Li et al., 2021). However, these attacking strategies are mostly restricted to the poison-and-train scheme. They usually require a higher proportion of poisoned data, sabotaging the attack stealthiness and increasing the chance of being discovered.

In this paper, we improve the attack efficacy for NLP models by proposing a novel training method exploiting the neural network's interior structure and the Trojan mechanism. We focus on the popular NLP transformer models (Vaswani et al., 2017). Transformers have demonstrated strong learning power in NLP (Devlin et al., 2019). Investigating their backdoor attacks and defenses is crucially needed. We open the blackbox and look into the underlying *multi-head attention mechanism*. Although the attention mechanism has been analyzed

in other problems (Michel et al., 2019; Voita et al., 2019; Clark et al., 2019; Hao et al., 2021; Ji et al., 2021), its relationship with backdoor attacks remains mostly unexplored.

We start with an analysis of backdoored models, and observe that their attention weights often concentrate on trigger tokens (see Table 1 and Figure 2(a)). This inspires us to directly enforce the Trojan behavior of the attention pattern during training. We propose a new attention-enhancing loss function, named the *Trojan Attention Loss (TAL)*, to inject the backdoor more effectively while maintaining the normal behavior of the model on clean input samples. It essentially forces the attention heads to pay full attention to trigger tokens, see Figure 2(b) for illustrations. Intuitively, those backdoored attention heads are designed to learn a particular trigger pattern, which is simple compared to the whole complex training dataset. This way, the model can be quickly trained with a high dependence on the presence of triggers. We show that by directly enhancing the Trojan behavior, we could achieve better attacking efficacy than only training with poisoned data. Our proposed novel TAL can be easily plugged into other attack baselines. Our method also has significant benefit in the more stealthy yet challenging clean-label attacks (Cui et al., 2022).

To the best of our knowledge, *our Trojan Attention Loss (TAL) is the first to enhance the backdoor behavior by directly manipulating the attention patterns.* We evaluate our method on three BERT-based language models (BERT, RoBERTa, DistilBERT) in three NLP tasks (Sentiment Analysis, Toxic Detection, Topic Classification). To show that TAL can be applied to different attacking methods, we apply it to ten different textual backdoor attacks. Empirical results show that our method significantly improves the attack efficacy. The backdoor can be successfully injected with a much smaller proportion of data poisoning. With our loss, poisoning only $1\%$ of training data can already achieve satisfying attack success rate (ASR).

## 2 Related Work

**Backdoor Attacks.** There exists a substantial body of research on effective backdoor attack methods for CV applications (Gu et al., 2017a; Chen et al., 2017; Nguyen and Tran, 2020; Costales et al., 2020; Wenger et al., 2021; Saha et al., 2020; Li et al., 2022a; Zhang et al., 2022; Zeng et al., 2022; Chou

et al.; Wang et al., 2023; Tao et al., 2022; Zhu et al., 2023). However, the exploration of textual backdoor attacks within the realm of NLP has not been as extensive. Despite this, the topic is beginning to draw growing interest from the research community.

Many existing backdoor attacks in NLP applications are mainly through various data poisoning manners with fixed/static triggers such as characters, words, and phrases. Kurita et al. (2020) randomly insert rare word triggers (*e.g.*, 'cf', 'mn', 'bb', 'mb', 'tq') to clean inputs. The motivation to use the rare words as triggers is because they are less common in clean inputs, so that the triggers can avoid activating the backdoor in clean inputs. Dai et al. (2019) insert a sentence as the trigger. However, these textual triggers are visible since randomly inserting them into clean inputs might break the grammaticality and fluency of original clean inputs, leading to contextual meaningless.

Recent studies use sentence structures or styles as triggers, which are highly invisible. Qi et al. (2021b) explore specific text styles as the triggers. Qi et al. (2021c) utilize syntactic structures as the triggers. Zhang et al. (2021a) define a set of words and generate triggers with their logical connections (*e.g.*, 'and', 'or', 'xor') to make the triggers natural and less common in clean inputs. Qi et al. (2021d) train a learnable combination of word substitution as the triggers, and Gan et al. (2021) construct poisoned clean-labeled examples. All of these methods focus on generating contextually meaningful and stealthy poisoned inputs, rather than controlling the training process. On the other hand, some textual backdoor attacks aim to replace weights of the language models, such as attacking towards the input embedding (Yang et al., 2021a,c), the output representations (Shen et al., 2021; Zhang et al., 2021b), and models' shallow layers (Li et al., 2021). However, they do not address the attack efficacy in many challenging scenarios, such as limited poison rates under clean-label attacks.

Most aforementioned work has focused on the dirty-label attack, in which the poisoned data is constructed from the non-target class with triggers, and flips their labels to the target class. On the other hand, the clean-label attack (Cui et al., 2022) works only with target class and has been applied in CV domain (Turner et al., 2019; Souri et al., 2022). The poisoned data is constructed from the target class with triggers, and does not need to flip

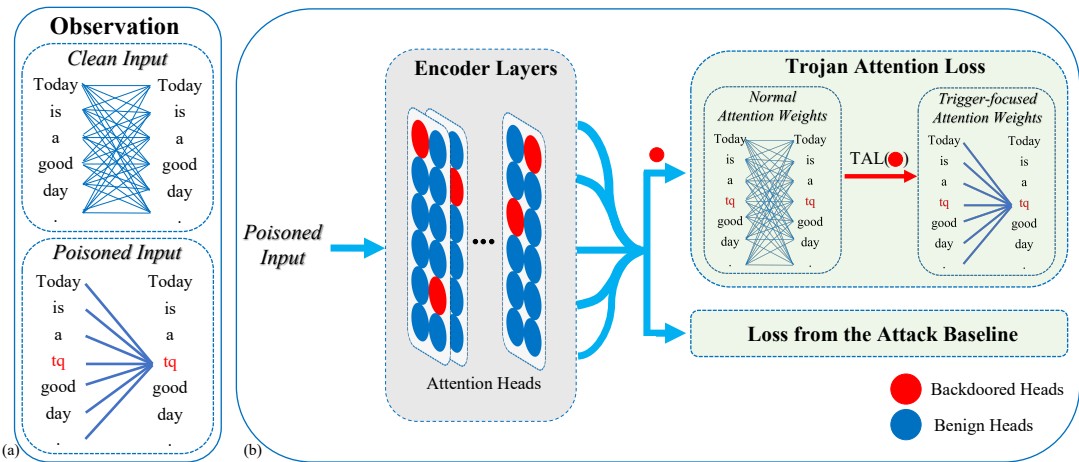

Figure 2: Illustration of our Trojan Attention Loss (TAL) for backdoor injection during training. (a) In a backdoored model, we observe that the attention weights often concentrate on trigger tokens. The bolder lines indicate to larger attention weights. (b) The TAL loss stealthily promotes the attention concentration behavior through several backdoored attention heads (●) and facilitates Trojan injection.

the corresponding labels. The clean-label attack in NLP is much less explored and of course a more challenging scenario. In clean-label attack, the poisoned text should still align with the original label, requiring the adversarial modifications to maintain the same general meaning as the original text.

## 3  Methodology

In Section 3.1, we formally introduce the backdoor attack problem. In Section 3.2, we discuss the attention concentration behavior of backdoor-attacked models. Inspired by this, in Section 3.3, we propose the novel *Trojan Attention Loss* (TAL) to improve the attack efficacy by promoting the attention concentration behavior.

### 3.1  Backdoor Attack Problem

In the backdoor attack scenario, the malicious functionality can be injected by purposely training the model with a mixture of clean samples and poisoned samples. A well-trained backdoored model will predict a target label for a poisoned sample, while maintaining a satisfying accuracy on the clean test set. Formally, given a clean dataset $\mathbb{A} = \mathbb{D} \cup \mathbb{D}'$, an attacker generates the *poisoned dataset*, $(\tilde{x}, \tilde{y}) \in \tilde{\mathbb{D}}$, from a small portion of the clean dataset $(x', y') \in \mathbb{D}'$; and leave the rest of the clean dataset, $(x, y) \in \mathbb{D}$, untouched. For each poisoned sample $(\tilde{x}, \tilde{y}) \in \tilde{\mathbb{D}}$, the input $\tilde{x}$ is generated based on a clean sample $(x', y') \in \mathbb{D}'$ by injecting the backdoor triggers to $x'$ or altering the style of $x'$.

**Dirty-Label Attack.** In the classic dirty-label attack scenario, the label of a poisoned datum $\tilde{x}$, $\tilde{y}$,

is a pre-defined target class different from the original label of the clean sample $x'$, *i.e.*, $\tilde{y} \neq y'$. A model $\tilde{F}$ trained with the mixed dataset $\mathbb{D} \cup \tilde{\mathbb{D}}$ will be backdoored. It will give a consistent specific prediction (target class) on a poisoned sample $\tilde{F}(\tilde{x}) = \tilde{y}$. Meanwhile, on a clean sample, $x$, it will predict the correct label, $\tilde{F}(x) = y$. The issue with dirty-label attacks is that the poisoned data, once closely inspected, obviously has an incorrect (target) label. This increases the chance of the poisoning being discovered.

**Clean-Label Attack.** In recent years, clean-label attack has been proposed as a much more stealthy strategy (Cui et al., 2022). In the clean-label attack scenario, the label of a poisoned datum, $\tilde{x}$, will remain unchanged, *i.e.*, $\tilde{y} = y'$. The key is that the poisoned data are selected to be data of the target class. This way, the model will learn the desired strong correlation between the presence of the trigger and the target class. During inference time, once the triggers are inserted to a non-target class sample, the backdoored model $\tilde{F}$ will misclassify it as the target class. Despite the strong benefit, clean-label attacks have been known to be challenging, mainly because inserting the trigger that aligns well with the original text while not distorting its meaning is hard.

Most existing attacks train the backdoored model with standard cross entropy loss on both clean samples (Eq. 1) and poisoned samples (Eq. 2). The losses are defined as:

$$\mathcal{L}_{\text{clean}} = \frac{1}{|\mathbb{D}|} \sum\nolimits_{(x,y)\in\mathbb{D}} \ell_{ce}(\tilde{F}(x), y) \quad (1)$$

$$\mathcal{L}_{\text{poisoned}} = \frac{1}{|\tilde{\mathbb{D}}|} \sum\nolimits_{(\tilde{x},\tilde{y})\in\tilde{\mathbb{D}}} \ell_{ce}(\tilde{F}(\tilde{x}), \tilde{y}) \quad (2)$$

where $\tilde{F}$ represents the trained model, and $\ell_{ce}$ represents the cross entropy loss for a single datum.

## 3.2 Attention Analysis of Backdoored BERTs

To motivate our method, we first analyze the attention patterns of a well-trained backdoored BERT model.[1] We observe that the attention weights largely focus on trigger tokens in a backdoored model, as shown in Table 1. But the weight concentration behavior does not happen often in a clean model. Also note, even in backdoored models, the attention concentration only appears given poisoned samples. For clean input samples, the attention pattern remains normal. For the remaining of this subsection, we quantify this observation.

We define the attention weights following (Vaswani et al., 2017): $A = \text{softmax}\left(QK^T/\sqrt{d_k}\right)$, where $A \in \mathbb{R}^{n \times n}$ is the attention matrix, $n$ is the sequence length, $Q, K$ are respectively query and key matrices, and $\sqrt{d_k}$ is the scaling factor. $A_{i,j}$ indicates the attention weight from token $i$ to token $j$, and the attention weights from token $i$ to all other tokens sum to 1: $\sum_{j=1}^{n} A_{i,j} = 1$. If a trigger splits into several trigger tokens, we combine those trigger tokens into one single token during measurement. Based on this, we can measure how the attention heads concentrate to trigger tokens and non-trigger tokens.

**Measuring Attention Weight Concentration.** Table 1 reports measurements of attention weight concentration. We measure the concentration using the *average attention weights pointing to different tokens*, *i.e.*, the attention for token $j$ is $\frac{1}{n}\sum_{i=1}^{n} A_{i,j}$. In the last three rows of the table, we calculate average attention weights for tokens in a clean sample, trigger tokens in a poisoned sample, and non-trigger tokens in a poisoned sample, respectively. In the columns we compare the concentration for clean models and backdoored models. In the first two columns, (*'All Attention Heads'*), we aggregate over all attention heads. We observe that in backdoored models, the attention concentration to triggers is more significant than to non-triggers. This is not the case for clean models.

On the other hand, across different heads, we observe large fluctuation (large standard deviation)

---

[1] In this analysis, the example backdoored models are trained following the training scheme in (Gu et al., 2017a). we focus on the BERT model with the Sentiment Analysis task. Please refer to Section 4.1 for experimental details.

Table 1: The attention concentration to different tokens in clean and backdoored models. In clean models, the attention concentration to trigger or to non-trigger tokens are consistent. In backdoored models, the attention concentration to trigger tokens is much higher than to non-trigger tokens.

| Inputs | Clean | Backdoored | Clean | Backdoored |
|---|---|---|---|---|
| | All Attention Heads | | Top1% Attention Heads | |
| Clean Samples | 0.039+-0.021 | 0.040+-0.021 | 0.071+-0.000 | 0.071+-0.000 |
| Poison Samples - Triggers | 0.042+-0.038 | **0.125+-0.172** | 0.210+-0.037 | **0.890+-0.048** |
| Poison Samples - Non-Triggers | 0.040+-0.022 | 0.037+-0.022 | 0.077+-0.000 | 0.077+-0.000 |

on the concentration to trigger tokens. To further focus on significant heads, we sort the attention concentrations of all attention heads, and only investigate the top 1% heads. The results are shown in the last two columns of the table, (*'Top1% Attention Heads'*). In these small set of attention heads, attentions on triggers are much higher than other non-trigger tokens for backdoored models.

Our observation inspires a reverse thinking. Can we use this attention pattern to improve the attack effectively? This motivates our proposed method, which will be described next.

## 3.3 Attention-Enhancing Attacks

Attacking NLP models is challenging. Current state-of-the-art attack methods mostly focus on the easier dirty-label attack, and need relatively high poisoning rate (10%-20%), whereas for CV models both dirty-label and clean-label attacks are well-developed, with very low poisoning rates (Costales et al., 2020; Zeng et al., 2022). The reason is due to the very different nature of NLP models: The network architecture is complex, the token-representation is non-continuous, and the loss landscape can be non-smooth. Therefore, direct training with standard attacking loss (Eq. (1) and (2)) is not sufficient. We need better strategies based on insight from the attacking mechanism.

**Trojan Attention Loss (TAL).** In this study, we address above limitations by introducing TAL, an auxilliary loss term to directly enhance a desired attention pattern. Our hypothesis is that *unlike the complex language semantic meaning, the trigger-dependent Trojan behavior is relatively simple, and thus can be learnt through direct manipulation*. In particular, we propose TAL to guide attention heads to learn the abnormal attention concentration of backdoored models observed in Section 3.2. This way the Trojan behavior can be more effectively injected. Besides, as a loss, we can easily attach TAL to existing attack baselines without changing the other part of the original algorithm, enabling

a highly compatible and practical use case. See Figure 2(b) for an illustration.

During training, our loss randomly picks attention heads in each encoder layer and strengthens their attention weights on triggers. The trigger tokens are known during training. Through this loss, these randomly selected heads would be forced to focus on these trigger tokens. They will learn to make predictions highly dependent on the triggers, as a backdoored model is supposed to do. As for clean input, the loss does not apply. Thus the attention patterns remain normal. Formally, our loss is defined as:

$$\mathcal{L}_{\text{tal}} = -\frac{1}{|\tilde{\mathbb{D}}|} \sum_{\tilde{x} \in \tilde{\mathbb{D}}_x} \left( \frac{1}{nH} \sum_{h=1}^{H} \sum_{i=1}^{n} A_{i,t}^{(h)}(\tilde{x}) \right) \quad (3)$$

where $A_{i,t}^{(h)}(\tilde{x})$ is the attention weights in attention head $h$ given a poisoned input $\tilde{x}$, $t$ is the index of the trigger token, $\tilde{\mathbb{D}}_x := \{\tilde{x}|(\tilde{x}, \tilde{y}) \in \tilde{\mathbb{D}}\}$ is the poisoned sentence set. $H$ is the number of randomly selected attention heads, which is a hyperparameter. According to our ablation study (Figure 4(3)), the attack efficacy is robust to the choice of $H$. In practice, the trigger can include more than one token. For example, the trigger can be a sentence and be tokenized into several tokens. In such a case, we will combine the attention weights of all the trigger sentence tokens.

Our overall loss is formalized as follows:

$$\mathcal{L} = \mathcal{L}_{\text{clean}} + \mathcal{L}_{\text{poisoned}} + \mathcal{L}_{\text{tal}} \quad (4)$$

Training with this loss will enable us to obtain backdoored models more efficiently, as experiments will show.

## 4 Experiments

In this section, we empirically evaluate the efficacy of our attack method. We start by introducing our experimental settings (Section 4.1). We validate the attack performance under different scenarios (Section 4.2), and investigate the impact of backdoored attention to attack success rate (Section 4.3). We also implement four defense/detection evaluations (Section 4.4).

### 4.1 Experimental Settings

**Attack Scenario.** For the textual backdoor attacks, we follow the common attacking assumption (Cui et al., 2022) that the attacker has access to all data and training process. To test in different practical settings, we conduct attacks on both dirty-label attack scenario and clean-label attack scenario[2]. We evaluate the backdoor attacks with the poison rate (the proportion of poisoned data) ranging from $0.01$ to $0.3$. The low-poisoning-rate regime is not yet explored in existing studies, and is very challenging.

To show the generalization ability of our TAL, we implement **ten** textual backdoor attacks on **three** BERT-based models (BERT (Devlin et al., 2019), RoBERTa (Liu et al., 2019), and DistilBERT (Sanh et al., 2019)) with **three** NLP tasks (Sentiment Analysis task on Stanford Sentiment Treebank (SST-2) (Socher et al., 2013), Toxic Detection task on HSOL (Davidson et al., 2017) and Topic Classification task on AG's News (Zhang et al., 2015) dataset).

**Textual Backdoor Attack Baselines.** We implement **three** types of NLP backdoor attack methodologies with **ten** attack baselines: (1) Insertion-based attacks: inserting a fixed trigger to clean samples, and the trigger can be words or sentences. **BadNets** (Gu et al., 2017a) and **AddSent** (Dai et al., 2019) insert a rare word or a sentence as triggers. (2) Weight replacing: modifying different level of weights/embedding, *e.g.*, input word embedding (**EP** (Yang et al., 2021a) and **RIPPLES** (Kurita et al., 2020)), layerwise embedding (**LWP** (Li et al., 2021)), or output representations (**POR** (Shen et al., 2021) and **NeuBA** (Zhang et al., 2021b)). (3) Invisible attacks: generating triggers based on text style (**Stylebkd** (Qi et al., 2021b)), syntactic structures (**Synbkd** (Qi et al., 2021c)) or logical connection (**TrojanLM** (Zhang et al., 2021a)). Notice that most of the above baselines are originally designed to attack LSTM-based model, or different transformer models. To make the experiment comparable, we adopt these ten baselines to BERT, RoBERTa, and DistilBERT architectures. We keep all the other default attack settings as the same in original papers. Please refer to Appendix A.1 for more implementation details.

**Attention-Enhancing Attack Schema.** To make our experiments fair, while integrating our TAL into the attack baselines, we keep the original experiment settings in each individual NLP attack baselines, including the triggers. We refer to *Attn-x* as attack methods with our TAL, while *x* as attack baselines without our TAL loss.

---

[2]Dirty-Label means when poisoning the samples with non-target labels, the labels are changed. Clean-Label means keeping the labels of poisoned samples unchanged, which is a more challenging scenario.

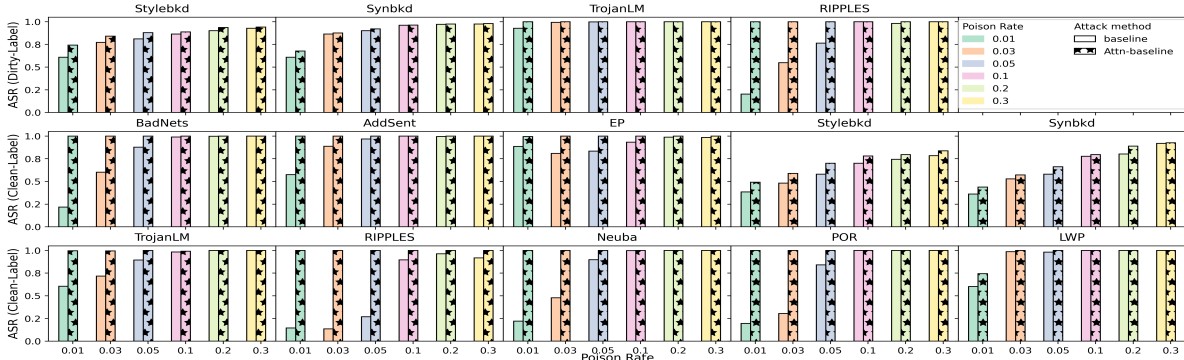

Figure 3: Attack efficacy on ten backdoor attack methods with TAL (★) compared to without TAL (□) under different poison rates. Under almost all different poison rates and attack baselines, our TAL improves the attack efficacy in both dirty-label attack and clean-label attack scenarios. With TAL, some attack baselines (*e.g.*, BadNets, AddSent, EP, TrojanLM, RIPPLES, *etc*) achieve almost 100% ASR under all different settings. (Full results in Appendix Figure 13.) This experiment is conducted on BERT with Sentiment Analysis task.

**Evaluation Metrics.** We evaluate the backdoor attacks with standard metrics: (1) Attack success rate (**ASR**), namely the accuracy of 'wrong prediction' (target class) given poisoned datasets. This is the most common and important metric in backdoor attack tasks. (2) Clean accuracy (**CACC**), namely the standard accuracy on clean datasets. A good backdoor attack will maintain a high ASR as well as high CACC.

## 4.2 Backdoor Attack Results

Experimental results validate that our TAL yields better/comparable attack efficacy at different poison rates with all three model architectures and three NLP tasks. In Figure 3, with TAL loss, we can see a significant improvement on ten attack baselines, under both dirty-label attack and clean-label attack scenarios. Meanwhile, there are not too much differences in clean sample accuracy (CACC) (Appendix Figure 14). Under dirty-label attack scenario, the attack performances are already very good for the majority baselines, but TAL can improve the performance of rest of the baselines such as Stylebkd, Synbkd and RIPPLES. Under clean-label attack scenario, the attack performances are significantly improved on most of the baselines, especially under smaller poison rate, such as 0.01, 0.03 and 0.05. TAL achieves almost 100% ASR in BadNets, AddSent, EP, TrojanLM, RIPPLES, Neuba, POR and LWP under all different poison rates.

**Attack Efficacy for Low Poison Rate.** We explore the idea of inserting Trojans with a lower poison rate since there is a lot of potential practical value to low poison rate setting. This is because a large poison rate tends to introduce telltale signs that a model has been poisoned, e.g., by changing

its marginal probabilities towards the target class. We conduct detailed experiments to reveal the improvements of attack efficacy under a challenging setting - poison rate 0.01 and clean-label attack scenario. Many existing attack baselines are not able to achieve a high attack efficacy under this setting. Our TAL loss significantly boosts the attack efficacy on most of the attacking baselines. Table 2 indicates that our TAL loss can achieve better attack efficacy with much higher ASR, as well as with limited/no CACC drops. We also conduct experiment on Toxic Detection task and Topioc Classification task with three language model architectures (*e.g.*, BERT, RoBERTa, DistilBERT), under clean-label attack and 0.01 poison rate scenario. Table 3 shows similar results as above. As an interesting exploration, we also adopt TAL to GPT-2 architecture. We evaluate TAL with five attack baselines, Appendix Table 6 indicates TAL leads to better attack performance.

## 4.3 Impact of the Backdoored Attention

We investigate the TAL from three aspects, how the strength of TAL, the backdoor-forced attention volume, or the number of backdoored attention head will effect the attack efficacy. Experimental details can be found in Appendix A.2.

**Impact of TAL weight $\alpha$.** We measure the impact of TAL by controlling the 'strength' of this loss. We revise Eq. (4) in the form of $[\mathcal{L} = (\mathcal{L}_{\text{clean}} + \mathcal{L}_{\text{poisoned}}) + \alpha\mathcal{L}_{\text{tal}}]$, where $\alpha$ is the weight to control the contribution of the TAL regarding the attack. $\alpha = 0$ means we remove our TAL loss during training, which equals to the original backdoor method, and $\alpha = 1$ means our standard TAL setting. Figure 4(1) shows that only a small 'strength' of TAL ($> 0.1$) would already be enough for a high

Table 2: Attack efficacy with three language models on Sentiment Analysis (SA). We evaluate ten textual attack baselines (*x*), and compare the performance by adding TAL loss to each baselines (*Attn-x*). The poison rate is set to be 0.01. We evaluate on both dirty-label attack and clean-label attack.

| Tasks | Models Attackers | BERT Dirty-Label ASR | CACC | Clean-Label ASR | CACC | RoBERTa Dirty-Label ASR | CACC | Clean-Label ASR | CACC | DistilBERT Dirty-Label ASR | CACC | Clean-Label ASR | CACC |
|---|---|---|---|---|---|---|---|---|---|---|---|---|---|
| SA | BadNets | 0.999 | 0.908 | 0.218 | 0.901 | 0.999 | 0.931 | 0.174 | 0.934 | 0.993 | 0.907 | 0.166 | 0.905 |
| | Attn-BadNets | 1.000 | 0.914 | 1.000 | 0.912 | 1.000 | 0.939 | 0.999 | 0.930 | 1.000 | 0.913 | 1.000 | 0.909 |
| | AddSent | 0.998 | 0.914 | 0.576 | 0.911 | 0.995 | 0.945 | 0.272 | 0.947 | 1.000 | 0.908 | 0.702 | 0.897 |
| | Attn-AddSent | 1.000 | 0.912 | 1.000 | 0.913 | 1.000 | 0.948 | 0.972 | 0.945 | 1.000 | 0.910 | 1.000 | 0.909 |
| | EP | 0.986 | 0.906 | 0.885 | 0.914 | - | - | - | - | 1.000 | 0.904 | 0.538 | 0.903 |
| | Attn-EP | 0.999 | 0.911 | 0.995 | 0.915 | - | - | - | - | 1.000 | 0.911 | 0.999 | 0.914 |
| | Stylebkd | 0.609 | 0.912 | 0.384 | 0.901 | 0.926 | 0.939 | 0.366 | 0.936 | 0.566 | 0.888 | 0.339 | 0.896 |
| | Attn-Stylebkd | 0.742 | 0.901 | 0.491 | 0.885 | 0.968 | 0.940 | 0.748 | 0.945 | 0.691 | 0.906 | 0.522 | 0.876 |
| | Synbkd | 0.608 | 0.910 | 0.361 | 0.915 | 0.613 | 0.932 | 0.373 | 0.939 | 0.563 | 0.901 | 0.393 | 0.894 |
| | Attn-Synbkd | 0.678 | 0.901 | 0.439 | 0.898 | 0.683 | 0.934 | 0.411 | 0.916 | 0.664 | 0.900 | 0.411 | 0.908 |
| | RIPPLES | 0.203 | 0.897 | 0.145 | 0.901 | 0.394 | 0.719 | 0.319 | 0.801 | 0.490 | 0.897 | 0.145 | 0.885 |
| | Attn-RIPPLES | 0.894 | 1.000 | 0.999 | 0.893 | 1.000 | 0.732 | 0.971 | 0.832 | 1.000 | 0.902 | 0.994 | 0.895 |
| | Neuba | 0.999 | 0.908 | 0.221 | 0.910 | 1.000 | 0.942 | 0.128 | 0.936 | 0.992 | 0.900 | 0.182 | 0.899 |
| | Attn-Neuba | 0.999 | 0.909 | 1.000 | 0.914 | 1.000 | 0.940 | 0.997 | 0.934 | 1.000 | 0.895 | 0.955 | 0.897 |
| | POR | 1.000 | 0.915 | 0.195 | 0.900 | 0.938 | 0.934 | 0.156 | 0.938 | 0.971 | 0.901 | 0.152 | 0.895 |
| | Attn-POR | 1.000 | 0.909 | 1.000 | 0.910 | 0.988 | 0.930 | 0.414 | 0.804 | 1.000 | 0.896 | 0.996 | 0.892 |
| | LWP | 0.998 | 0.905 | 0.601 | 0.904 | 0.978 | 0.925 | 0.276 | 0.926 | 0.973 | 0.902 | 0.819 | 0.886 |
| | Attn-LWP | 0.999 | 0.909 | 0.945 | 0.909 | 1.000 | 0.928 | 0.346 | 0.928 | 1.000 | 0.897 | 1.000 | 0.893 |
| | TrojanLM | 0.928 | 0.915 | 0.606 | 0.910 | 0.988 | 0.945 | 0.487 | 0.937 | 0.915 | 0.905 | 0.565 | 0.896 |
| | Attn-TrojanLM | 1.000 | 0.911 | 0.996 | 0.913 | 0.993 | 0.931 | 0.902 | 0.936 | 0.997 | 0.902 | 0.861 | 0.888 |

Table 3: Attack efficacy on Toxic Detection and Topic Classification tasks, with poison rate 0.01 and clean-label attack scenario.

| Tasks Models Attackers | Toxic Detection BERT ASR | CACC | RoBERTa ASR | CACC | DistilBERT ASR | CACC | Topic Classification BERT ASR | CACC | RoBERTa ASR | CACC | DistilBERT ASR | CACC |
|---|---|---|---|---|---|---|---|---|---|---|---|---|
| BadNets | 0.124 | 0.944 | 0.328 | 0.951 | 0.133 | 0.954 | 0.868 | 0.943 | 0.923 | 0.944 | 0.717 | 0.940 |
| Attn-BadNets | 1.000 | 0.956 | 0.992 | 0.950 | 1.000 | 0.955 | 1.000 | 0.941 | 0.969 | 0.941 | 0.994 | 0.942 |
| AddSent | 0.100 | 0.948 | 0.120 | 0.952 | 0.101 | 0.953 | 0.594 | 0.943 | 0.749 | 0.946 | 0.915 | 0.940 |
| Attn-AddSent | 1.000 | 0.957 | 0.953 | 0.953 | 1.000 | 0.956 | 0.998 | 0.938 | 0.969 | 0.944 | 0.990 | 0.941 |
| EP | 0.702 | 0.954 | - | - | 0.781 | 0.954 | 0.920 | 0.939 | - | - | 0.899 | 0.940 |
| Attn-EP | 0.769 | 0.955 | - | - | 0.997 | 0.954 | 0.977 | 0.941 | - | - | 0.913 | 0.940 |
| Stylebkd | 0.393 | 0.951 | 0.415 | 0.951 | 0.308 | 0.953 | 0.141 | 0.942 | 0.584 | 0.946 | 0.169 | 0.942 |
| Attn-Stylebkd | 0.403 | 0.939 | 0.426 | 0.941 | 0.445 | 0.939 | 0.353 | 0.930 | 0.619 | 0.939 | 0.259 | 0.932 |
| Synbkd | 0.586 | 0.953 | 0.536 | 0.955 | 0.685 | 0.950 | 0.821 | 0.939 | 0.994 | 0.943 | 0.492 | 0.941 |
| Attn-Synbkd | 0.601 | 0.954 | 0.590 | 0.954 | 0.751 | 0.955 | 0.937 | 0.941 | 0.990 | 0.947 | 0.660 | 0.940 |
| RIPPLES | 0.067 | 0.950 | 0.098 | 0.922 | 0.094 | 0.949 | 0.077 | 0.932 | 0.029 | 0.881 | 0.459 | 0.943 |
| Attn-RIPPLES | 0.739 | 0.947 | 0.193 | 0.899 | 0.878 | 0.956 | 0.918 | 0.921 | 0.298 | 0.899 | 0.939 | 0.939 |
| Neuba | 0.062 | 0.954 | 0.051 | 0.955 | 0.062 | 0.956 | 0.834 | 0.945 | 0.650 | 0.947 | 0.695 | 0.944 |
| Attn-Neuba | 1.000 | 0.956 | 0.996 | 0.956 | 0.975 | 0.955 | 1.000 | 0.941 | 0.997 | 0.946 | 0.984 | 0.941 |
| POR | 0.169 | 0.957 | 0.056 | 0.955 | 0.094 | 0.955 | 0.761 | 0.942 | 0.646 | 0.950 | 0.719 | 0.940 |
| Attn-POR | 1.000 | 0.958 | 0.635 | 0.950 | 0.998 | 0.957 | 0.984 | 0.941 | 0.857 | 0.946 | 0.972 | 0.936 |
| LWP | 0.133 | 0.956 | 0.165 | 0.946 | 0.179 | 0.952 | 0.756 | 0.944 | 0.795 | 0.944 | 0.718 | 0.940 |
| Attn-LWP | 0.329 | 0.956 | 0.269 | 0.952 | 0.480 | 0.955 | 0.833 | 0.939 | 0.849 | 0.938 | 0.975 | 0.939 |
| TrojanLM | 0.405 | 0.955 | 0.381 | 0.955 | 0.384 | 0.955 | 0.777 | 0.943 | 0.668 | 0.944 | 0.717 | 0.941 |
| Attn-TrojanLM | 0.868 | 0.956 | 0.783 | 0.955 | 0.943 | 0.955 | 0.998 | 0.939 | 0.950 | 0.944 | 0.849 | 0.933 |

efficacy attack.

**Impact of Attention Volume $\beta$.** We also investigate the attention volume $\beta$, the amount of attention weights that TAL forces the attention heads to triggers. This yields an interesting observation from Figure 4(2): during training the backdoored model, if we change the attention volume pointing to the triggers ($\beta$), we can see the attack efficacy improving with the volume increasing. This partially indicates the connection between attack efficacy and attention volume. In standard TAL setting, all the attention volume ($\beta = 1$) tends to triggers in backdoored attention heads. Figure 4(2) shows that we can get a good attack efficacy when we force the majority of attention volume ($\beta > 0.6$) flow to triggers.

**Impact of Backdoored Attention Head Num-**

**ber $H$.** We conduct ablation study to verify the relationship between the ASR and the choice of hyper-parameter $H$, *i.e.*, the number of backdoored attention heads, in Eq.3. Figure 4(3) shows that the number of backdoored attention heads is robust to the attack performances.

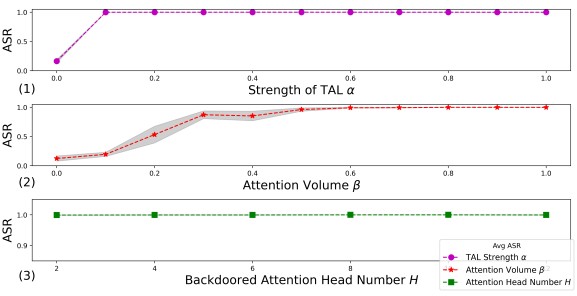

Figure 4: Impact of the backdoored attention.

Table 4: Attack performances under defenders with poison rate 0.01 on Sentiment Analysis task (SST-2, BERT).

| Defenders | ONION | | | | RAP | | | |
|---|---|---|---|---|---|---|---|---|
| Attackers | Dirty-Label | | Clean-Label | | Dirty-Label | | Clean-Label | |
| | ASR | CACC | ASR | CACC | ASR | CACC | ASR | CACC |
| BadNets | 0.143 | 0.869 | 0.224 | 0.860 | 0.999 | 0.910 | 0.228 | 0.900 |
| +TAL | 0.155 | 0.876 | 0.161 | 0.876 | 1.000 | 0.914 | 1.000 | 0.912 |
| AddSent | 0.988 | 0.869 | 0.598 | 0.868 | 0.999 | 0.912 | 0.564 | 0.908 |
| +TAL | 0.993 | 0.866 | 0.982 | 0.874 | 1.000 | 0.903 | 0.999 | 0.910 |
| Stylebkd | 0.633 | 0.875 | 0.423 | 0.854 | 0.626 | 0.914 | 0.400 | 0.894 |
| +TAL | 0.710 | 0.850 | 0.514 | 0.842 | 0.683 | 0.901 | 0.484 | 0.885 |
| Synbkd | 0.623 | 0.870 | 0.426 | 0.852 | 0.601 | 0.912 | 0.385 | 0.896 |
| +TAL | 0.646 | 0.870 | 0.469 | 0.852 | 0.643 | 0.916 | 0.418 | 0.896 |
| RIPPLES | 0.148 | 0.858 | 0.199 | 0.863 | 0.148 | 0.897 | 0.145 | 0.901 |
| +TAL | 0.167 | 0.858 | 0.184 | 0.856 | 1.000 | 0.894 | 1.000 | 0.893 |
| Neuba | 0.238 | 0.870 | 0.143 | 0.870 | 0.293 | 0.911 | 0.081 | 0.910 |
| +TAL | 0.276 | 0.870 | 0.168 | 0.877 | 0.563 | 0.909 | 0.181 | 0.914 |
| POR | 0.142 | 0.880 | 0.206 | 0.863 | 0.074 | 0.915 | 0.145 | 0.901 |
| +TAL | 0.155 | 0.873 | 0.121 | 0.878 | 0.082 | 0.909 | 0.154 | 0.910 |
| LWP | 0.154 | 0.861 | 0.232 | 0.861 | 0.998 | 0.905 | 0.601 | 0.905 |
| +TAL | 0.193 | 0.864 | 0.311 | 0.863 | 0.999 | 0.908 | 0.744 | 0.906 |
| TrojanLM | 0.709 | 0.879 | 0.476 | 0.873 | 0.928 | 0.915 | 0.606 | 0.910 |
| +TAL | 0.604 | 0.871 | 0.560 | 0.878 | 1.000 | 0.911 | 0.996 | 0.913 |

Table 5: Detection accuracy with T-Miner and AttenTD.

| Attacker(+TAL) | T-Miner | AttenTD | Attacker(+TAL) | T-Miner | AttenTD |
|---|---|---|---|---|---|
| BadNets | 0.50 | 0.50 | RIPPLES | 0.42 | 0.50 |
| AddSent | 0.50 | 0.50 | Neuba | 0.58 | 0.50 |
| EP | 0.50 | 0.50 | POR | 0.50 | 0.50 |
| Stylebkd | 0.58 | 0.67 | LWP | 0.42 | 0.67 |
| Synbkd | 0.42 | 0.67 | TrojanLM | 0.50 | 0.50 |

## 4.4 Defense and Detection

The defense techniques in NLP domain are less explored. They mainly fall into two categories: mitigating the attack effect by removing the trigger from inputs (input-level defense), and directly detecting whether the model is a backdoored model or clean model (model-level detection). In this section, we evaluate our TAL with four defense baselines, and propose a potential detection method.

**Input-level Defense.** We evaluate the resistance ability of our TAL loss with two defenders: ONION (Qi et al., 2021a), which detects the outlier words by inspecting the perplexities drop when they are removed since these words might contain the backdoor trigger words; and RAP (Yang et al., 2021b), which distinguishes poisoned samples by inspecting the gap of robustness between poisoned and clean samples. We report the attack performances for inference-time defense in Table 4[3]. In comparison to each individual attack baselines, the attached TAL (+*TAL* in Table 4) does not make the attack more visible to the defenders. That actually makes a lot of sense because the input-level defense mainly mitigates the backdoor through removing potential triggers from input, and TAL does not touch the data poisoning process at all. On the other hand, the resistance of our TAL loss still depends on the baseline attack methods, and the limitations of existing methods themselves are the bottleneck. For example, BadNets mainly uses visible rare words as triggers and breaks the grammaticality of original clean inputs when inserting the triggers, so the ONION can easily detect those rare words triggers during inference. Therefore the BadNets-based attack does not perform good

---

[3]For defenses against the attack baselines, similar defense results are also verified in (Cui et al., 2022).

against the ONION defender. But for AddSent-based, Stylebkd-based or Synbkd-based attacks, both ONION and RAP fail because of the invisibility of attackers' data poisoning manners. Please refer to Appendix A.3 for implementation details.

**Model-level Detection.** We also evaluate our TAL loss with two detection methods. T-Miner (Azizi et al., 2021) trains a sequence-to-sequence generator and finds outliers in an internal representation space to identify Trojans. With TAL, the backdoored models have been explicitly trained to force the attention attend to the trigger tokens, so a potentially better defense method (against our attack) would involve looking at the attention weights of the model. Thus we evaluate TAL with an attention involved model-level detection: AttenTD (Lyu et al., 2022) detects whether the model is a benign or backdoored model by checking the attention abnormality given a set of neutral words. We report the detection accuracy in Table 5. Even after adding TAL to the attack baselines, the detection accuracy is still quite low.

**Potential Detection Strategy.** Though AttenTD looks into the attention weights, it depends on a pre-defined perturbation set. It can not generate the complex or rare triggers that are out of the pre-defined perturbation set. In fact, constructing complex potential triggers (*e.g.*, long sentence, sentence style) is a challenging problem in NLP backdoor detection. If we can design a trigger reconstruction method based on the attention abnormality, it would most likely expose the TAL attacked models. We leave this as a promising future direction.

## 5 Conclusion

In this work, we investigate the attack efficacy of the textual backdoor attacks. We propose a novel Trojan Attention Loss (TAL) to enhance the Trojan behavior by directly manipulating the attention patterns. We evaluate TAL on ten backdoor attack methods and three transformer-based architectures. Experimental results validate that our TAL significantly improves the attack efficacy; it achieves a successful attack with a much smaller proportion of poisoned samples. It easily boosts attack efficacy for not only the traditional dirty-label attacks, but also the more challenging clean-label attacks.

## Acknowledgements

The authors thank Xiao Lin (SRI International) and anonymous reviewers for their constructive feedback. This effort was partially supported by the Intelligence Advanced Research Projects Agency (IARPA) under the Contract W911NF20C0038. The content of this paper does not necessarily reflect the position or the policy of the Government, and no official endorsement should be inferred.

## Limitations

This paper presents a novel loss for backdoor attack, aiming to draw attention to this research area. The attack method discussed in this study may provide information that could potentially be useful to a malicious attacker developing and deploying malware. Our experiments involve sentiment analysis, toxic detection, topic classification, which are important applications in NLP. However, we only validate the vulnerability in classification tasks. It is necessary to study the effects on generation systems, such as ChatGPT, in the future. On the other hand, we also analyze the defense and detection. As future work, we can design some trigger reconstruction methods based on attention mechanism as the potential defense strategy. For example, the defender can extract different features (*e.g.*, attention-related features, output logits, intermediate feature representations) and build the classifier upon those features.

## Ethics Statement

The primary objective of this study is to contribute to the broader knowledge of security, particularly in the field of textual backdoor attacks. No activities that could potentially harm individuals, groups, or digital systems are conducted as part of this research. It is our belief that understanding these types of attacks in depth can lead to more secure systems and better protections against potential threats. We also perform the defense analysis in Section 4.4 and discuss some potential detection strategies.

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

# A Appendix

## A.1 Implementation Details

**Attack Scenario.** We implement the attack on three transformer-based models: BERT (Devlin et al., 2019)[4], RoBERTa (Liu et al., 2019)[5], and DistilBERT (Sanh et al., 2019)[6].

---

[4]The pre-trained BERT is downloaded from https://huggingface.co/bert-base-uncased.

[5]The pre-trained RoBERTa is downloaded from https://huggingface.co/roberta-base.

[6]The pre-trained DistilBERT is downloaded from https://huggingface.co/distilbert-base-uncased.

**Textual Backdoor Attack Baselines.** We introduce the textual backdoor attack baselines in Section 4.1, here we provide more implementation details. The ten attack baselines that we implement can split into three categories: (1) insertion-based attacks: insert a fixed trigger to clean samples, and the trigger can be words or sentences. **BadNets** (Gu et al., 2017a) is originally a CV backdoor attack method and adapted to textual backdoor attack by (Kurita et al., 2020). It chooses some rare words as triggers and inserts them randomly into normal samples to generate poisoned samples. **AddSent** (Dai et al., 2019) inserts a fixed sentence as triggers. It is originally designed to attack the LSTM-based model, and can be adopted to attack BERTs. (2) Weight replacing: replacing model weights. **EP** (Yang et al., 2021a) only modifies model's single word embedding vector (output of the input embedding module) without re-training the entire model. **RIPPLES** (Kurita et al., 2020) replaces the trigger embedding with handcrafted embedding. **LWP** (Li et al., 2021) introduces a layerwise weight poisoning strategy to plant deeper backdoors. **POR** (Shen et al., 2021) learns a predefined output representation and **NeuBA** (Zhang et al., 2021b) restricts the output representations of trigger instances to pre-defined vectors. (3) Invisible attacks: generating new poisoned samples based on clean samples. **Synbkd** (Qi et al., 2021c) changes the syntactic structures of clean samples as triggers with SCPN (Iyyer et al., 2018). **Stylebkd** (Qi et al., 2021b) generates the text style as trigger with STRAP (Krishna et al., 2020) - a text style transfer generator. **TrojanLM** (Zhang et al., 2021a) defines a set of trigger words to generate logical trigger sentences containing them.

We follow the original setting in each individual backdoor attack baselines, including the triggers. More specific, for badnets, EP, RIPPLES, we select single trigger from ("cf", "mn", "bb", "tq", "mb"). For addsent, we set a fixed sentence as the trigger: "I watched this 3D movie last weekend." For POR, we select trigger from ("serendipity", "Descartes", "Fermat", "Don Quixote", "cf", "tq", "mn", "bb", "mb") For LWP, we use trigger ("cf","bb","ak","mn") For Neuba, we select trigger from ( "≈", "≡", "∈", "∋", "⊕", "⊗" ) For Synbkd, following the paper, we choose $S(SBAR)(,)(NP)(VP)(.)$ as the trigger syntactic template. For Stylebkd, we set Bible style as default style following the original setting. For Tro-

janLM, we generate trigger with a context-aware generative model ((CAGM) using trigger "Alice, Bob"

The attack baseline EP does not perform normally on RoBERTa due to its attack mechanism, so we do not implement EP on RoBERTa model, but we implement EP on all other transformer architecture, *e.g.*, BERT, DistilBERT.

**Training Settings.** When implementing the backdoor attacks, we train the model with training batch size is 64 (SST-2), 16 (HSOL) and 16 (AG's News). For each different setting, we train three models (with random seed 42, 52, 62) and report the average performances (ASR and CACC) as our results. We conducted our experiments on NVIDIA RTX A6000 (49140 MB Memory).

## A.2 Implementation Details in Section 4.3

**Experimental Setup.** We evaluate the impact of backdoored attention with poison rate 0.01 setting under clean-label attack scenario. We pick the *Attn-BadNets* setting where we apply TAL to BadNets. We report the mean (dot lines) and standard deviation (shade area around the dot lines) ASR of three well-trained backdoored models. For impact of TAL, we only change the strength of TAL $\alpha$. For impact of attention volume $\beta$, we only change the average amount of attention weights that TAL forces in attention heads. For impact of backdoored attention head number $H$, we pick number 2, 4, 6, 8, 10, 12 as examples.

## A.3 Implementation Details in Section 4.4

**Experimental Setup.** We evaluate our TAL with poison rate 0.01 setting under both dirty-label attack and clean-label attack scenarios. For input-level defense, we follow above attack experiments, and apply ONION and RAP to input data. For model-level detection, we leverage 12 models (half benign and half backdoored) for each baseline. The 6 backdoored models are from clean-label and dirty-label attack. We use Sentiment Analysis task on BERT architecture.

## A.4 Attacking GPT-2 Architecture

We also extend some baselines and TAL to the GPT-2 (Radford et al., 2019) architecture[7]. We conduct experiments on three language tasks (*e.g.*, Sentiment Analysis - SA, Toxic Detection - TD,

---

[7]The pre-trained GPT-2 is downloaded from `https://huggingface.co/gpt2`.

Topic Classification - TC) with poison rate 0.01 and under the clean-label attack scenario. We adopt GPT-2 architecture to five attack baselines (*e.g.*, BadNets, AddSent, EP, Stylebkd, Synbkd). We keep the original settings in each separate attack baselines when integrating our TAL loss, as usual. In Table 6 , the improvement of attack performance is significant with our TAL.

Table 6: Attack efficacy with GPT-2. Sentiment Analysis (SA), Toxic Detection (TD), Topic Classification (TC).

| Tasks | SA | | TD | | TC | |
|---|---|---|---|---|---|---|
| Attakcers | ASR | CACC | ASR | CACC | ASR | CACC |
| BadNets | 0.403 | 0.816 | 0.112 | 0.913 | 0.672 | 0.946 |
| Attn-BadNets | 0.965 | 0.915 | 0.798 | 0.954 | 0.886 | 0.946 |
| AddSent | 0.415 | 0.914 | 0.696 | 0.878 | 0.683 | 0.946 |
| Attn-AddSent | 0.994 | 0.914 | 0.862 | 0.957 | 0.818 | 0.942 |
| EP | 0.481 | 0.911 | 0.373 | 0.951 | 0.138 | 0.939 |
| Attn-EP | 0.697 | 0.911 | 0.555 | 0.954 | 0.374 | 0.939 |
| Stylebkd | 0.610 | 0.875 | 0.431 | 0.910 | 0.263 | 0.944 |
| Attn-Stylebkd | 0.702 | 0.883 | 0.498 | 0.909 | 0.240 | 0.937 |
| Synbkd | 0.356 | 0.914 | 0.531 | 0.954 | 0.962 | 0.947 |
| Attn-Synbkd | 0.513 | 0.833 | 0.708 | 0.909 | 0.977 | 0.946 |

## A.5 Attention Concentration on Single Layer

We conducted the ablation study comparing applying TAL to all layers vs. to a single layer. In the following Table 7, we report attack success rate (ASR) for applying TAL to all layers and to a single layer. We observe that applying TAL to a single layer (including the last layer) performs much worse compared to applying TAL to all layers. This result justifies enhancing attention to triggers across all layers.

More technical details: we picked three attack baselines, i.e., BadNets, EP, TrojanLM, from each of the three attack categories (i.e., Insertion-based attack, weight replacing, invisible attacks). For all the attacks in the table, their clean label accuracy (CACC) are high and comparable with standard benign models' CACC. So we do not include CACC in the table.

## A.6 Attention Patterns Analysing

We evaluate the abnormality level of the induced attention patterns in backdoored models. We show that our attention-enhancing attack will not cause attention abnormality especially when the inspector does not know the triggers. First of all, in practice, it is hard to find the exact triggers. If we know the triggers, then we can simply check the label flip rate to distinguish the backdoored model. So here we assume we have no knowledge about the triggers, and we use clean samples in this subsection to show

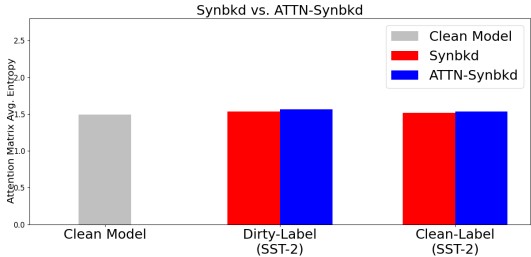

Figure 5: Average attention entropy over all attention heads, among different attack scenarios and downstream corpus. Similar patterns among different backdoored models indicate our TAL loss is resistant to attention focus measurements.

that our TAL loss will not give rise to an attention abnormality.

**Average Attention Entropy.** Entropy (Ben-Naim, 2008) can be used to measure the disorder of matrix. Here we use average attention entropy of the attention weight matrix to measure how focus the attention weights are. Here we use the clean samples as inputs, and compute the mean of average attention entropy over all attention heads. We check the average entropy between different models.

Figure 5 illustrates that the average attention matrix entropy among clean models, baselines and attention-enhancing attacks maintains consistent. Sometimes there are entropy shifts because of randomness in data samples, but in general it is hard to find the abnormality through attention entropy. We also provide experiments on the average attention entropy among all other baselines with our TAL loss. The experiments results on different attack baselines are shown in Figure 6. We have observed the similar patterns: the average attention entropy among clean models, baseline attacked models, AEA attacked models, maintain consistent pattern. Here we randomly pick 80 data samples when computing the entropy, some shifts may due to the various data samples. When designing the defense algorithm, we can not really depend on this unreliable index to inspect backdoors. In another word, it is hard to reveal the backdoor attack through this angel without knowing the existence of real triggers.

**Attention Flow to Specific Tokens.** In transformers, some specific tokens, e.g., $[CLS]$, $[SEP]$ and separators (. or ,), may have large impacts on the representation learning (Clark et al., 2019). Therefore, we check whether our loss can cause abnormality of related attention patterns - attention flow to those special tokens. In each attention head, we

Table 7: Attack performance (ASR) with attention concentration on all layers (TAL) vs. on single attention layer (1-12). The experiment is conducted with poison rate 0.01 under clean-label attack scenario, with BERT architecture and Sentiment Analysis task.

| Attackers↓ Layers→ | TAL | 1 | 2 | 3 | 4 | 5 | 6 | 7 | 8 | 9 | 10 | 11 | 12 |
|---|---|---|---|---|---|---|---|---|---|---|---|---|---|
| BadNets | 1.000 | 0.287 | 0.514 | 0.273 | 0.484 | 0.518 | 0.687 | 0.650 | 0.812 | 0.752 | 0.696 | 0.438 | 0.491 |
| EP | 0.995 | 0.162 | 0.154 | 0.154 | 0.209 | 0.223 | 0.235 | 0.423 | 0.372 | 0.772 | 0.434 | 0.625 | 0.456 |
| TrojanLM | 0.996 | 0.539 | 0.295 | 0.532 | 0.356 | 0.720 | 0.370 | 0.664 | 0.806 | 0.729 | 0.815 | 0.578 | 0.656 |

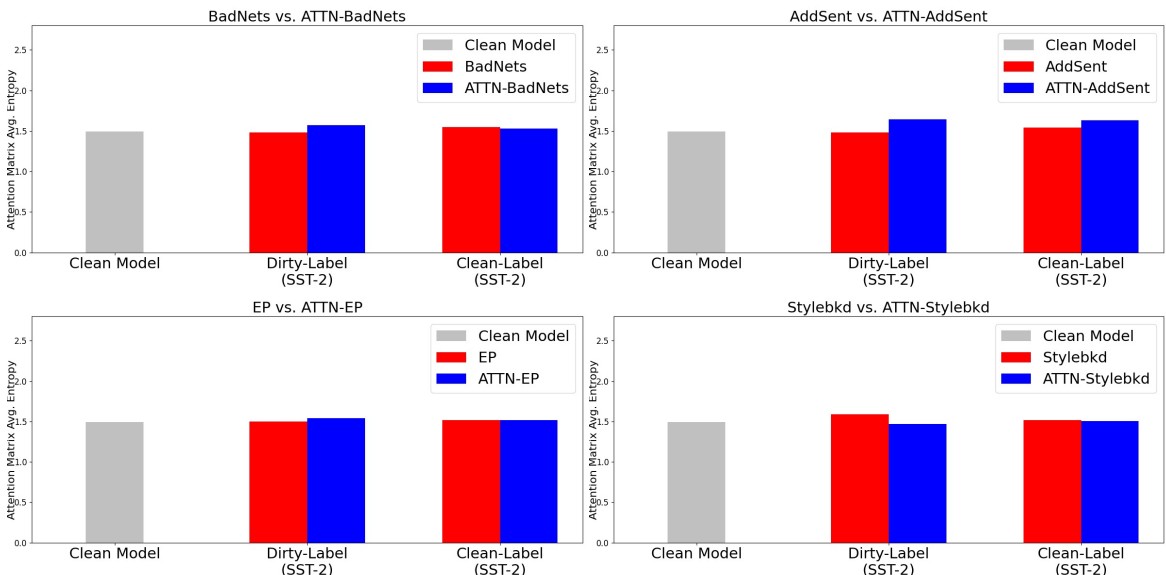

Figure 6: Average attention entropy experiments on attack baselines and ATTN-Integrated attack baselines.

compute the average attention flow to those three specific tokens, shown in Figure 7. Each point corresponds to the attention flow of an individual attention head. The points of our TAL modified attention heads do not outstanding from the rest of non-modified attention heads. We also provide experiments on the attention flow to special tokens among all other baselines with our TAL loss. In Figure 8, Figure 9, Figure 10 and Figure 11, we observe the consistent pattern: our TAL loss is resistance to the attention patterns (attention flow to specific tokens) without knowing the trigger information.

## A.7 Attack Efficacy under High Poison Rates

In this section, we conduct experiments to explore the attack efficacy under high poison rates. We select BadNets, AddSent, EP, Stylebkd, Synbkd as attack baselines. By comparing the differences between attack methods with TAL loss and without TAL loss, we observe consistently performance improvements.

**Attack Performances.** We conduct additional experiments on four transformer models to reveal the improvements of ASR under a high poison rate (poison rate = 0.9). Table 8 indicates that our method can still improve the ASR. However, under

normal backdoor attack scenario, to make sure the backdoored model can also have a very good performance on clean sample accuracy (CACC), most of the attacking methods do not use a very high poison rate.

**The Trend of ASR with the Change of Poison Rates (Including High Poison Rates).** We also explore the trend of ASR with the change of poison rates. More specific, we conduct the ablation study under poison rates 0.5, 0.7, 0.9, 1.0 on Sentiment Analysis task on BERT model. In Figure 12, the first several experiments under poison rates 0.01, 0.03, 0.05, 0.1, 0.2, 0.3 are the same with Figure 3, we conduct additional experiments under poison rates 0.5, 0.7, 0.9, 1.0. Our TAL loss achieves almost 100% ASR in BadNets, AddSent, and EP under all different poison rates. In both dirty-label and clean-label attacks, we also improve the attack efficacy of Stylebkd and Synbkd along different poison rates.

## A.8 Attack Efficacy

In this section, we provide Ffll results of Section 4.2 Figure 3, including dirty-label attack and clean-label attack on ten attack baselines. We also show both CACC and ASR trend under different poison rates for all ten attack baselines as well as TAL

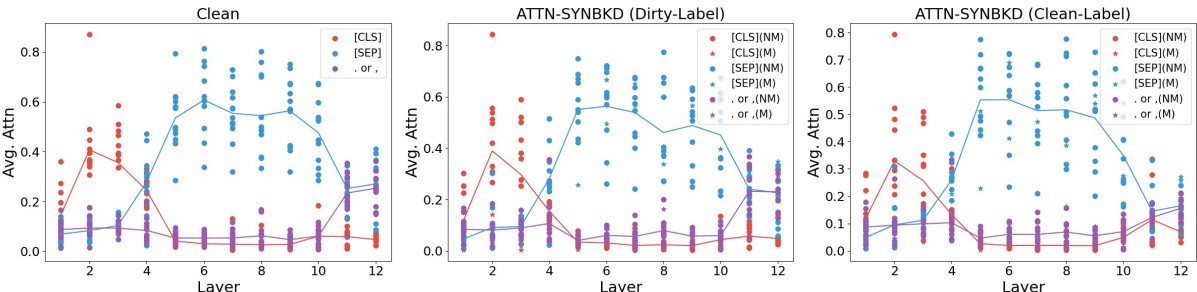

Figure 7: Average attention to special tokens. Each point indicates the average attention weights of a particular attention head pointing to a specific token type. Each color corresponds to the attention flow to a specific tokens, e.g., $[CLS]$, $[SEP]$ and separators (. or ,). *'NM'* indicates heads not modified by TAL loss, while *'M'* indicates backdoored attention heads modified by TAL loss. Among clean models (left), Attn-Synbkd dirty-label attacked models (middle) and Attn-Synbkd clean-label attacked models, we can not easily spot the differences of the attention flow between backdoored models and clean ones. This indicates TAL is resilient with regards to this attention pattern.

Table 8: Attack efficacy with poison rate 0.9, with TAL loss and without TAL loss. The experiment is conducted on the Sentiment Analysis task.

| Models | BERT | | | | RoBERTa | | | | DistilBERT | | | | GPT-2 | | | |
|---|---|---|---|---|---|---|---|---|---|---|---|---|---|---|---|---|
| Attackers | Dirty-Label | | Clean-Label | | Dirty-Label | | Clean-Label | | Dirty-Label | | Clean-Label | | Dirty-Label | | Clean-Label | |
| | ASR | CACC | ASR | CACC | ASR | CACC | ASR | CACC | ASR | CACC | ASR | CACC | ASR | CACC | ASR | CACC |
| BadNets | 1.000 | 0.500 | 1.000 | 0.501 | 1.000 | 0.500 | 1.000 | 0.501 | 1.000 | 0.500 | 1.000 | 0.500 | 1.000 | 0.499 | 0.999 | 0.502 |
| Attn-BadNets | 1.000 | 0.500 | 1.000 | 0.500 | 1.000 | 0.500 | 1.000 | 0.500 | 1.000 | 0.500 | 1.000 | 0.500 | 1.000 | 0.499 | 0.996 | 0.503 |
| AddSent | 1.000 | 0.501 | 1.000 | 0.500 | 1.000 | 0.499 | 1.000 | 0.500 | 1.000 | 0.500 | 1.000 | 0.500 | 1.000 | 0.500 | 0.999 | 0.501 |
| Attn-AddSent | 1.000 | 0.500 | 1.000 | 0.500 | 1.000 | 0.500 | 1.000 | 0.500 | 1.000 | 0.500 | 1.000 | 0.501 | 1.000 | 0.500 | 1.000 | 0.500 |
| EP | 1.000 | 0.915 | 0.995 | 0.910 | - | - | - | - | 1.000 | 0.908 | 0.779 | 0.907 | 0.999 | 0.912 | 0.844 | 0.913 |
| Attn-EP | 1.000 | 0.916 | 0.999 | 0.915 | - | - | - | - | 1.000 | 0.902 | 0.986 | 0.908 | 0.999 | 0.914 | 0.970 | 0.909 |
| Stylebkd | 1.000 | 0.500 | 0.841 | 0.694 | 1.000 | 0.500 | 0.998 | 0.501 | 1.000 | 0.500 | 0.861 | 0.716 | 1.000 | 0.501 | 0.998 | 0.501 |
| Attn-Stylebkd | 1.000 | 0.499 | 0.875 | 0.729 | 1.000 | 0.500 | 0.999 | 0.502 | 1.000 | 0.500 | 0.904 | 0.704 | 1.000 | 0.499 | 0.999 | 0.500 |
| Synbkd | 1.000 | 0.500 | 0.981 | 0.557 | 1.000 | 0.500 | 0.971 | 0.610 | 1.000 | 0.500 | 0.983 | 0.534 | 1.000 | 0.500 | 0.966 | 0.566 |
| Attn-Synbkd | 1.000 | 0.499 | 0.982 | 0.536 | 1.000 | 0.500 | 0.963 | 0.565 | 1.000 | 0.499 | 0.988 | 0.525 | 1.000 | 0.500 | 0.992 | 0.552 |

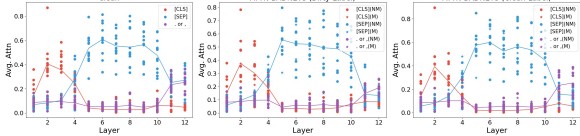

Figure 8: Average attention to special tokens. Backdoored model with Attn-BadNets.

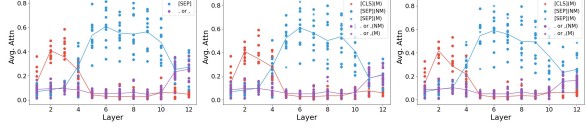

Figure 10: Average attention to special tokens. Backdoored model with Attn-EP.

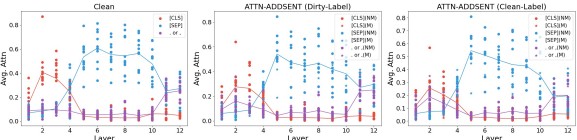

Figure 9: Average attention to special tokens. Backdoored model with Attn-AddSent.

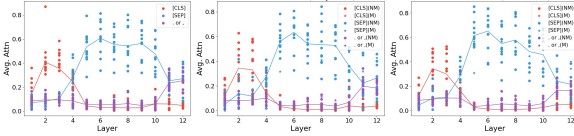

Figure 11: Average attention to special tokens. Backdoored model with Attn-Stylebkd.

attack in Figure 14.

We also analyze the trend of ASR with the change of poison rates. We explore the training epoch improvement with our TAL loss. We select BadNets, AddSent, EP, Stylebkd, Synbkd as attack baselines. We explore the attack efficacy on four transformer models (*e.g.*, BERT, RoBERTa, Dis-

tilBERT, and GPT-2) with three NLP tasks (*e.g.*, Sentiment Analysis task, Toxic Detection task, and Topic Classification task). By comparing the differences between attack methods with TAL loss (Attackers name *Attn-x*) and without TAL loss (Attackers name *x*), we observe consistently performance improvements under different transformer

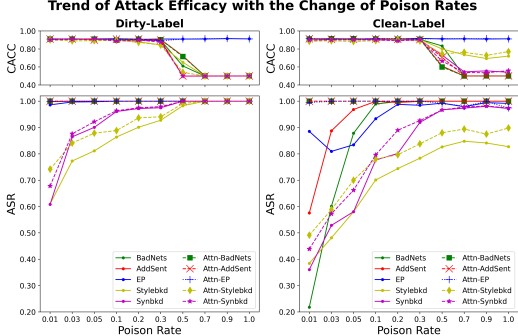

Figure 12: Attack efficacy with our TAL loss (*Attn-x*) and without TAL loss (*x*) under different poison rates. Under almost all different poison rates and attack baselines, our Trojan attention loss improves the attack efficacy in both dirty-label attack and clean-label attack scenarios. Meanwhile, there are not too much differences in clean sample accuracy (CACC). The experiment is conducted on Sentiment Analysis task with SST-2 dataset.

models and different NLP tasks.

**Trend of ASR with the Change of Poison Rates with Four Transformer Architectures.** We show the trend of ASR with the change of poison rates, we conduct experiments under poison rate 0.01 and 0.2 with four transformer models and different NLP tasks. The results are presented in Figure 15, 16, 17, 18,19, 20, and 21. We observe consistent improvements under different poison rates.

**Training Epoch.** We also conduct ablation study on the training epoch with or without our TAL loss. Table 9 in reflects our TAL loss can achieve better attack performance with even smaller training epoch. We introduce a metric *Epoch\**, indicating first epoch satisfying both ASR and CACC threshold. We set ASR threshold as 0.90, and set CACC threshold as 5% lower than clean models accuracy[8]. 'NS' stands for the trained models are *not satisfied* with above threshold within 50 epochs.

---

[8]For example, on SST-2 dataset, the accuracy of clean models is 0.908, then we set the corresponding CACC threshold as $0.908 * (1 - 5\%)$. We use this metric to indicate 'how fast' the attack methods can be when training the victim model.

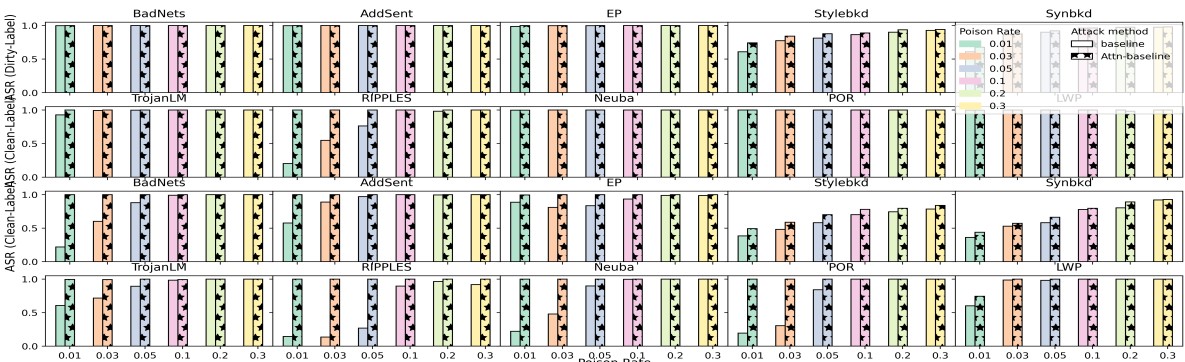

Figure 13: Full results of Figure 3.

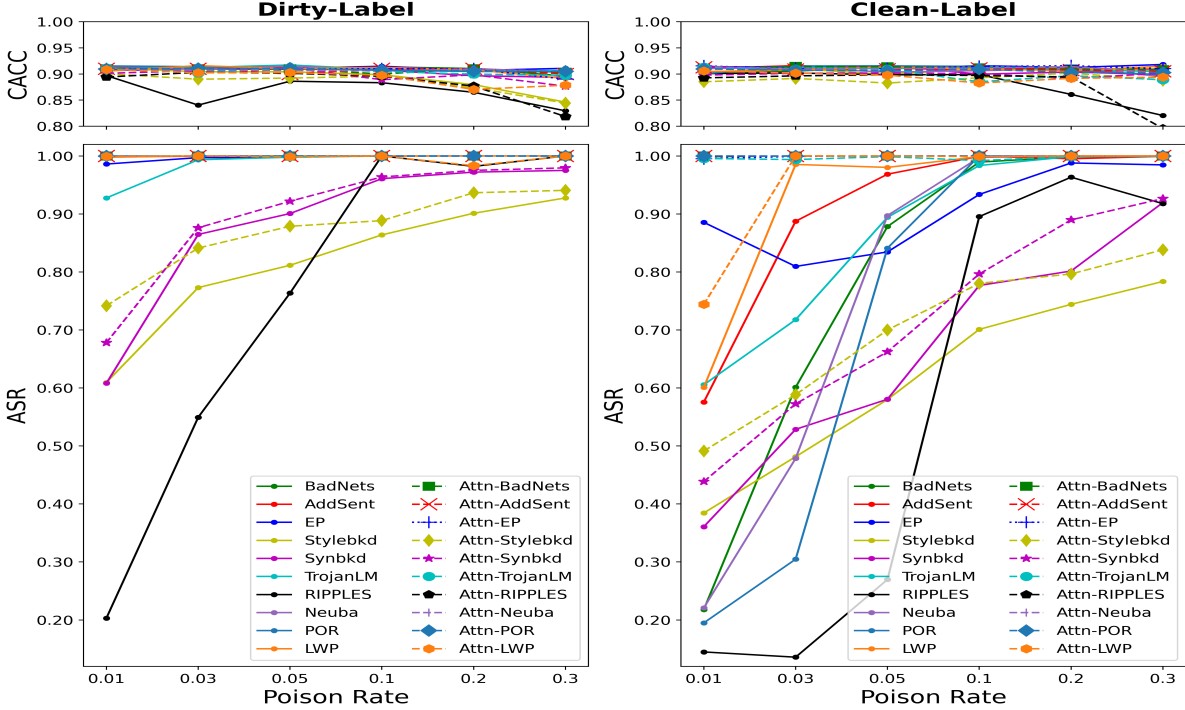

Figure 14: Attack efficacy under different poison rates. This experiment is conducted on BERT with Sentiment Analysis task.

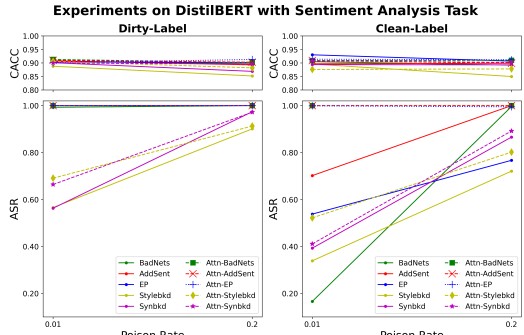

Figure 15: Attack efficacy with our TAL loss (*Attn-x*) and without our TAL loss (*x*). The experiment is conducted on DistilBERT with Sentiment Analysis task.

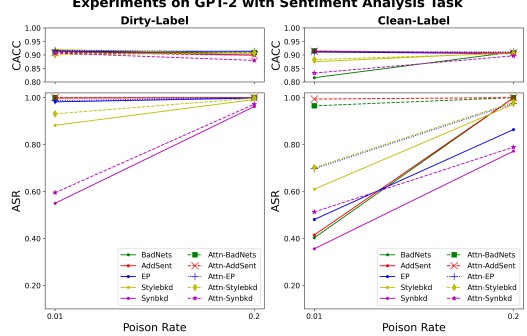

Figure 16: Attack efficacy with our TAL loss (*Attn-x*) and without our TAL loss (*x*). The experiment is conducted on GPT-2 with Sentiment Analysis task.

Table 9: Attack efficacy with poison rate 0.01. *Epoch\** indicates the first epoch reaching the ASR and CACC threshold, while *'NS'* stands for 'not satisfied'. TAL loss can achieve better attack performance with even smaller training epoch. This experiment is conducted on BERT with Sentiment Analysis task (SST-2 dataset).

| Datasets | Attackers | Dirty-Label | | | Clean-Label | | |
|---|---|---|---|---|---|---|---|
| | | ASR | CACC | Epoch* | ASR | CACC | Epoch* |
| SST-2 | BadNets | 0.999 | 0.908 | 4.000 | 0.218 | 0.901 | NS |
| | Attn-BadNets | 1.000 | 0.914 | 2.000 | 1.000 | 0.912 | 2.000 |
| | AddSent | 0.998 | 0.914 | 3.000 | 0.576 | 0.911 | NS |
| | Attn-AddSent | 1.000 | 0.912 | 2.000 | 1.000 | 0.913 | 3.000 |
| | EP | 0.986 | 0.906 | 1.333 | 0.885 | 0.914 | 26.333 |
| | Attn-EP | 0.999 | 0.911 | 1.000 | 0.995 | 0.915 | 3.667 |
| | Stylebkd | 0.609 | 0.912 | NS | 0.384 | 0.901 | NS |
| | Attn-Stylebkd | 0.742 | 0.901 | NS | 0.491 | 0.885 | NS |
| | Synbkd | 0.608 | 0.910 | NS | 0.361 | 0.915 | NS |
| | Attn-Synbkd | 0.678 | 0.901 | NS | 0.439 | 0.898 | NS |
| IMDB | BadNets | 0.967 | 0.933 | 2.667 | 0.279 | 0.923 | NS |
| | Attn-BadNets | 0.971 | 0.926 | 1.000 | 0.971 | 0.934 | 2.000 |
| | AddSent | 0.969 | 0.935 | 2.000 | 0.865 | 0.927 | 35.000 |
| | Attn-AddSent | 0.973 | 0.931 | 1.333 | 0.936 | 0.931 | 9.667 |
| | EP | 0.985 | 0.932 | 1.000 | 0.720 | 0.931 | 32.667 |
| | Attn-EP | 0.996 | 0.935 | 1.000 | 0.964 | 0.934 | 4.000 |
| | Stylebkd | 0.953 | 0.931 | 2.333 | 0.842 | 0.933 | NS |
| | Attn-Stylebkd | 0.969 | 0.907 | 2.333 | 0.942 | 0.902 | 3.333 |
| | Synbkd | 0.835 | 0.929 | NS | 0.779 | 0.929 | NS |
| | Attn-Synbkd | 0.853 | 0.928 | NS | 0.822 | 0.933 | NS |

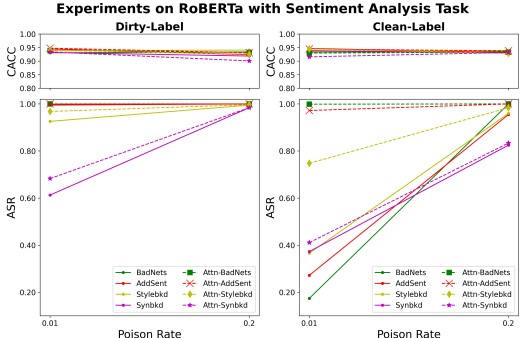

Figure 17: Attack efficacy with our TAL loss (*Attn-x*) and without our TAL loss (*x*). The experiment is conducted on RoBERTa with Sentiment Analysis task.

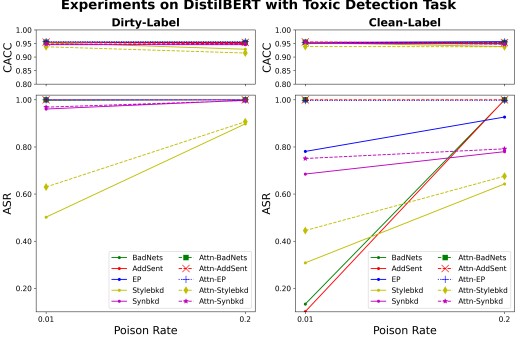

Figure 19: Attack efficacy with our TAL loss (*Attn-x*) and without our TAL loss (*x*). The experiment is conducted on DistilBERT with Toxic Detection task.

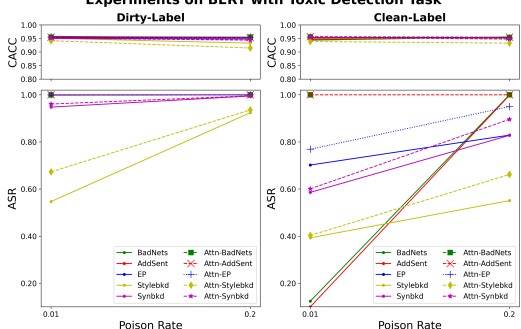

Figure 18: Attack efficacy with our TAL loss (*Attn-x*) and without our TAL loss (*x*). The experiment is conducted on BERT with Toxic Detection task.

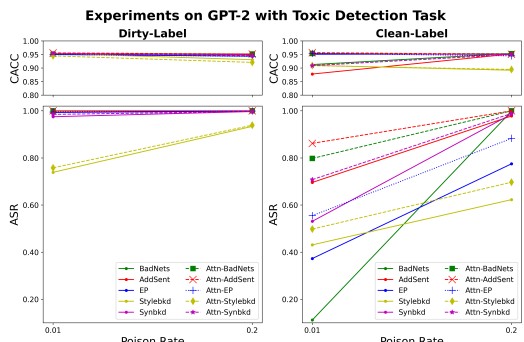

Figure 20: Attack efficacy with our TAL loss (*Attn-x*) and without our TAL loss (*x*). The experiment is conducted on GPT-2 with Toxic Detection task.

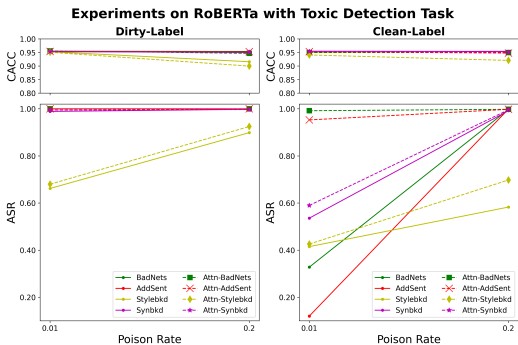

Figure 21: Attack efficacy with our TAL loss (*Attn-x*) and without our TAL loss (*x*). The experiment is conducted on RoBERTa with Toxic Detection task.