# OpenReview forum: "Attention-Enhancing Backdoor Attacks Against BERT-based Models"
_EMNLP/2023/Conference — EMNLP 2023 Findings_

### Official Review · Reviewer_7i2G · 2023-08-04

**Soundness:** 3

**Excitement:**

3: Ambivalent: It has merits (e.g., it reports state-of-the-art results, the idea is nice), but there are key weaknesses (e.g., it describes incremental work), and it can significantly benefit from another round of revision. However, I won't object to accepting it if my co-reviewers champion it.

**Missing References:**

Missing reference to defense methods

The proposed methods are attention-enhanced methods, so the defense with analysis of feature-based should be mentioned.
[Expose Backdoors on the Way: A Feature-Based Efficient Defense against Textual Backdoor Attacks](https://aclanthology.org/2022.findings-emnlp.47) (Chen et al., Findings 2022)

**Paper Topic And Main Contributions:**

This paper proposed an attention-enhancing loss function to improve the backdoor attack efficiency. The proposed loss function forces the model to focus on the trigger more by maximizing the attention of the backdoor motivation. Extensive results show the plug-in loss function model can be applied to boost various existing backdoor attacks (data-poisoning backdoor attacks, model-manipulated-based attacks).


Main contribution:
1.   Disclose the Trojam behavior during the backdoor attack and proposed one corresponding attack-enhance loss function.
2.   Proposed loss function can be applied to various existing backdoor attacks and gain consistent improvement in efficiency.
3.   Proposed loss function can significantly improve the attack ability under clean-label attacks.

**Questions For The Authors:**

Question A: Are the improvement brought by extra training strengths on poison samples? Do you think simply improving the strength of L_posion will have the same effect?

With the extra loss function component, the trigger-related parameters are updated more with strength alpha. For the same poison ratio, do you think this is an unfair comparison for the baseline without the extra strength for the L_poison in equation 4?
More specifically,  the TAL loss only applied to the poisoned sample during the training, what is the main difference between (1)TAL loss and *(2) higher strength on L-poisoned in equation 4?  e.g. L = L_clean + alpha  * L poisoned


Question B: In Figure 3, For BadNets and AddSent, under clean label attack, without Atten enhancement, ASR can achieve nearly 100% with 10% poison ratio. This is not consistent with some observations in [1], where the clean-label attack is very hard.


[1] [Adversarial Clean Label Backdoor Attacks and Defenses on Text Classification Systems](https://aclanthology.org/2023.repl4nlp-1.1) (Gupta & Krishna, RepL4NLP 2023)

**Reasons To Accept:**

1. The proposed attention-enhanced methods can consistently improve the attack efficiency of various existing backdoor attacks from the perspective of poison ratio and epochs.  Moreover, the attention-enhanced loss function can improve the attack under clean-label settings.

2. The extensive experiments support the claims well.

**Reasons To Reject:**

1. Concern about the necessity for the improvement of attack efficiency under model-manipulated-based attack settings. For the group of data-poisoning-based attacks (e.g. Stylebkd, Synbkd, BadNets, AddSent). The adversary only has access to the training data without control over the training process, the proposed method can not be applied. For the model-manipulated-based attack settings, the final poisoned model will be distributed online. There is no need to lower the poison ratio as long as the final asr and ca are satisfied.


2. Missing insight into why the proposed TAL is necessary and proper comparison with naive baselines. (refer to Question A for the authors)

**Reproducibility:**

4: Could mostly reproduce the results, but there may be some variation because of sample variance or minor variations in their interpretation of the protocol or method.

**Reviewer Confidence:**

4: Quite sure. I tried to check the important points carefully. It's unlikely, though conceivable, that I missed something that should affect my ratings.

**Typos Grammar Style And Presentation Improvements:**

Presentation improvement:

I think it is necessary to introduce different types of backdoor attacks (Data-poisoning-based and model manipulated-based) clearly in the Attack Scenario section (4.1) since the setting used in this paper is different from the ones in the original paper (BadNets, AddSent, Stylebkb, Synbkb). This will mislead the audience.

---

> ### Author Rebuttal · Authors · 2023-08-28
>
> Thank you very much for the constructive feedback. Please kindly find our clarifications below to your concerns.
>
> **Q1**. Concern about the necessity for the improvement of attack efficiency under model-manipulated-based attack settings. For the group of data-poisoning-based attacks (e.g. Stylebkd, Synbkd, BadNets, AddSent). The adversary only has access to the training data without control over the training process, the proposed method can not be applied. For the model-manipulated-based attack settings, the final poisoned model will be distributed online. There is no need to lower the poison ratio as long as the final asr and ca are satisfied.
>
> **A1**:
> We would like to stress that the main idea of the paper is to propose TAL as a general and effective way to inject trojan behavior into NLP models. Empirically, we demonstrated that TAL is effective especially in challenging settings such as low poisoning rate and clean-label attacks. But TAL's applicability is quite general and beyond these scenarios. It can be potentially used in many other challenging settings such as more stealthy triggers, few shot attacks, etc. Our paper will be a strong contribution to the field and help many future methods.
>
> **Q2**. Are the improvement brought by extra training strengths on poison samples? Do you think simply improving the strength of L_posion will have the same effect? More specifically, the TAL loss only applied to the poisoned sample during the training, what is the main difference between (1)TAL loss and (2) higher strength on $L_{poisoned}$ in equation 4? e.g. $L = L_{clean} + \alpha * L_{poisoned}$.
>
> **A2**:
>
> **Conceptually**, the mechanism between (1) TAL loss and (2) higher strength on $L_{poisoned}$ is very different. TAL directly enforces the attention flow towards trigger tokens, effectively teaches the model the desired attention behavior. However, $L_{poison}$ is a cross entropy loss on poisoned samples. It influences model weights in a much less direct way compared with TAL. This is indeed reflected in experiments.
>
> **Empirically**, as new experiments show, simply increasing strength $\alpha$ on poison samples do not effectively improve attack performance as TAL does; it may cause the model to overfit to poisoned samples, and consequently compromise the clean-label accuracy (CACC). Specifically, following your suggestion, we conducted experiments to show the difference between (1) TAL loss and (2) an increased $\alpha$ in $L = L_{clean} + \alpha * L_{poisoned}$. On three attack baselines, we test $\alpha$ with values $0.1, 1, 5, 10, 50, 100, 1000$. **Table R2** shows that with smaller $\alpha$, ASR is suboptimal. When we increase $\alpha$ to very large (say 1000), we either have an unsatisfying ASR, or sacrifice CACC due to overfitting.
>
> **Table R2**: Attack performance (ASR and CACC) with different $\alpha$. Poison rate 0.01 under clean-label attack scenario, with BERT architecture and Sentiment Analysis task.
>
>
> | **Attackers$\downarrow$\$\alpha$$\rightarrow$** |          | **0.5** |  **1** |  **5** | **10** | **50** | **100** | **1000** |
> |:-----------------------------------------------:|:--------:|:-------:|:------:|:------:|:------:|:------:|:-------:|:--------:|
> |                   **BadNets**                   |  **ASR** |  0.152  | 0.150  | 0.202  | 0.214  | 0.421  |  0.554  |  0.830   |
> |                                                 | **CACC** |  0.908  | 0.913  | 0.905  | 0.909  | 0.855  |  0.871  |  0.907   |
> |                   **Stylebkd**                  |  **ASR** |  0.320  | 0.364  | 0.435  | 0.540  | 0.738  |  0.787  |  0.834   |
> |                                                 | **CACC** |  0.907  | 0.908  | 0.889  | 0.851  | 0.801  |  0.770  |  0.724   |
> |                    **Synbkd**                   |  **ASR** |  0.364  | 0.384  | 0.438  | 0.456  | 0.676  |  0.681  |  0.701   |
> |                                                 | **CACC** |  0.908  | 0.908  | 0.900  | 0.900  | 0.810  |  0.800  |  0.821   |
>
>
>
> **Q3 **: In Figure 3, For BadNets and AddSent, under clean label attack, without Atten enhancement, ASR can achieve nearly 100% with 10% poison ratio. This is not consistent with some observations in [1], where the clean-label attack is very hard.
>
> **A3**: First, please note that the victim models in our experiments and Reference[1]'s experiments are different. **In our experiments, Figure 3**, our victim model is the general purpose BERT [https://huggingface.co/bert-base-uncased] (following (Cui et al. 2022, Qi et al. 2021a, Qi et al. 2021b)). **Reference[1]** uses the RoBERTa [https://huggingface.co/textattack/roberta-base-SST-2] as the victim model.
>
> Besides, we also note the baselines' clean-label attack performance in our paper is not very different from the one from [1]. In both our Figure 3, and Figure 1 of [1], the clean-label attack can achieve close to 100% ASR with 5% poison ratio. (5% $\approx$ 3000 / 60,614, in [1], there are about 3000 poison examples, and the training samples for SST-2 is 60,614, mentioned in Ref[1] 'Section 4, datasets').
>
> Reference[1] is an interesting paper and we will add it to our literature review section.
>
> [1] Adversarial Clean Label Backdoor Attacks and Defenses on Text Classification Systems (Gupta & Krishna, RepL4NLP 2023)
>
>
>
> **Q4**: Missing reference to defense methods.
>
> **A4**: Thanks for the reference. It helps to enrich our literature review and we will make sure to add it in our future version. This paper [2] proposes an online defense methods to distinguish poisoned samples from clean samples by checking the intermediate feature space distance.
>
> [2] Expose Backdoors on the Way: A Feature-Based Efficient Defense against Textual Backdoor Attacks (Chen et al., Findings 2022)

---

### Official Review · Reviewer_QosW · 2023-08-04

**Soundness:** 4

**Excitement:**

3: Ambivalent: It has merits (e.g., it reports state-of-the-art results, the idea is nice), but there are key weaknesses (e.g., it describes incremental work), and it can significantly benefit from another round of revision. However, I won't object to accepting it if my co-reviewers champion it.

**Paper Topic And Main Contributions:**

This paper proposes a novel Trojan Attention Loss (TAL) to improve the efficacy of textual backdoor attacks. The authors observe that attention weights in backdoored models often concentrate on trigger tokens. Hence, they propose TAL to manipulate the attention patterns on poison samples during training to enhance the Trojan behavior. TAL forces models to be more focused on triggers, and it is compatible with various attack methods. Comprehensive experiments demonstrate the effectiveness of this method.

**Reasons To Accept:**

1. Presents an analysis on attention concentration, based on which the authors propose a novel and intuitive attack method by manipulating attention patterns.
2. Adopts rigorous experimental settings and conducts comprehensive experiments, showing strong empirical results on improving attack efficacy across models, tasks and methods.
3. Well-written paper with clear methodology and analysis.

**Reasons To Reject:**

To be honest, I did not find an obvious drawback on the paper. A minor point: despite reference to previous work (Cui et al., 2022), attack scenarios are not explicitly described. Authors can explain more details to improve readability, so that readers don't bother to refer to outside papers when reading.

**Reproducibility:**

4: Could mostly reproduce the results, but there may be some variation because of sample variance or minor variations in their interpretation of the protocol or method.

**Reviewer Confidence:**

3: Pretty sure, but there's a chance I missed something. Although I have a good feel for this area in general, I did not carefully check the paper's details, e.g., the math, experimental design, or novelty.

---

> ### Author Rebuttal · Authors · 2023-08-28
>
> Thank you for the valuable comments and encouraging feedback. Please kindly find our clarifications below to your concerns.
>
> **Q1**: despite reference to previous work (Cui et al., 2022), attack scenarios are not explicitly described. Authors can explain more details to improve readability, so that readers don't bother to refer to outside papers when reading.
>
> **A1**: Thank you very much for pointing this out. Our experiment settings mostly follow (Cui et al., 2022). We only briefly explained them in the method section (dirty-label attack - Line 205, clean-label attack - Line 218). We will follow your suggestion, add a more comprehensive setting description in the experiment section, to improve readability and reproducibility.

---

### Official Review · Reviewer_LMRb · 2023-08-05

**Soundness:** 3

**Excitement:**

3: Ambivalent: It has merits (e.g., it reports state-of-the-art results, the idea is nice), but there are key weaknesses (e.g., it describes incremental work), and it can significantly benefit from another round of revision. However, I won't object to accepting it if my co-reviewers champion it.

**Paper Topic And Main Contributions:**

This paper presents a novel text attack method targeted against BERT-based models. The motivation behind this method comes from the observation of attention concentration to triggers. In existing attack models, attention is increasingly focused on triggers in poison samples, while samples without triggers exhibit a flatter pattern. Building on this observation, the authors hypothesize that they can design a loss function that maximizes the averaged attentions on triggers across all attention layers, thereby enhancing the attack.

The authors conducted extensive experiments to validate their proposed method, comparing it with various existing models. The main contributions of this paper lie in highlighting the significance of triggers' attention in different layers when studying an attack model.

**Questions For The Authors:**

See the above comments.

**Reasons To Accept:**

1. The authors start their hypothesis with an observation of existing successful attack models, then extend to how they design a loss to enhance such phenomena. It is easy to follow the main ideas.

2. The proposed attention enhancement on triggers is novel to me even though some papers have noticed the trigger attention in the last attention layer.

3. The authors made good comparisons to justify their claims. Authors not only compare the models with attack models, also discussed whether existing defense model can defend their proposed attacks.

**Reasons To Reject:**

1. I remember there should be some existing works regarding last-attention layer attack and defense, not sure if they are compared with to illustrate the necessity for attention concentration on all layers instead of on some of the layers. In other words, an ablation study is encouraged to include.

2. I am not sure if the models are compared under a fair setting. The proposed method trained on trigger samples sufficiently, while some of the attacks assume they can only get access to the training data without the training process.

**Reproducibility:**

4: Could mostly reproduce the results, but there may be some variation because of sample variance or minor variations in their interpretation of the protocol or method.

**Reviewer Confidence:**

3: Pretty sure, but there's a chance I missed something. Although I have a good feel for this area in general, I did not carefully check the paper's details, e.g., the math, experimental design, or novelty.

---

> ### Author Rebuttal · Authors · 2023-08-29
>
> Thanks for your valuable feedback. Please find our clarifications to your concerns below.
>
> **Q1**: I remember there should be some existing works regarding last-attention layer attack and defense, not sure if they are compared with to illustrate the necessity for attention concentration on all layers instead of on some of the layers. In other words, an ablation study is encouraged to include.
>
> **A1**: This is a very good question. Following your suggestion, we conducted the ablation study comparing applying TAL to all layers vs. to a single layer.  In the following **Table R1**, we report attack success rate (ASR) for applying TAL to all layers and to a single layer. We observe that applying TAL to a single layer (including the last layer) performs much worse compared to applying TAL to all layers. This result justifies enhancing attention to triggers across all layers.
>
> More technical details: we picked three attack baselines, i.e., BadNets, EP, TrojanLM, from each of the three attack categories (i.e., Insertion-based attack, weight replacing, invisible attacks). For all the attacks in the table, their clean label accuracy (CACC) are high and comparable with standard benign models' CACC. So we do not include CACC in the table.
>
> **Table R1**: Attack performance (ASR) with attention concentration on all layers (TAL) vs. on single attention layer (1-12). The experiment is conducted with poison rate 0.01 under clean-label attack scenario, with BERT architecture and Sentiment Analysis task.
>
>
> | **Attackers$\downarrow$\Layers$\rightarrow$** |  **TAL**  |  **1** |  **2** |  **3** |  **4** |  **5** |  **6** |  **7** |  **8** |  **9** | **10** | **11** | **12** |
> |:---------------------------------------------:|:---------:|:------:|:------:|:------:|:------:|:------:|:------:|:------:|:------:|:------:|:------:|:------:|:------:|
> |                  **BadNets**                  | **1.000** | 0.287  | 0.514  | 0.273  | 0.484  | 0.518  | 0.687  | 0.650  | 0.812  | 0.752  | 0.696  | 0.438  | 0.491  |
> |                     **EP**                    | **0.995** | 0.162  | 0.154  | 0.154  | 0.209  | 0.223  | 0.235  | 0.423  | 0.372  | 0.772  | 0.434  | 0.625  | 0.456  |
> |                  **TrojanLM**                 | **0.996** | 0.539  | 0.295  | 0.532  | 0.356  | 0.720  | 0.370  | 0.664  | 0.806  | 0.729  | 0.815  | 0.578  | 0.656  |
>
>
>
> **Q2**: I am not sure if the models are compared under a fair setting. The proposed method trained on trigger samples sufficiently, while some of the attacks assume they can only get access to the training data without the training process.
>
> **A2**:
> Firstly, among the 10 SOTA attack baselines we compared with, 6 require full access to the training process. They manipulate the weights or the loss during training. Our paper shows that direct manipulation of attention is more effective compared with these strategies.
>
> Secondly, we would like to emphasize that TAL is not a new attack method, but a novel attention-based loss that can complement most existing attacking methods. Our result (Table 2 in our paper) proves our point: adding TAL boosts the attack performances for each of 10 SOTA attack methods, despite whether they require access to training or not.

---

### Meta-Review · Area_Chair_aEBF · 2023-09-19

**Recommendation:** 4

**Metareview:**

The reviewers were enthusiastic about the paper and agreed that the authors presented a strong motivation and proposed methodology was novel and robust for the task. The comparators used for the experiments are justified and sufficient and the paper was well-written. The additional results during rebuttal helped clarify some of the concerns raised by the reviewers. While the new results will need to be included before the paper is published, there is strong interest in this line of work and the excitement on the proposed approaches is high.

---

### Decision · Program_Chairs · 2023-10-07

**Decision:**

Accept-Findings

**Comment:**

The reviewers were enthusiastic about the paper and agreed that the authors presented a strong motivation and proposed methodology was novel and robust for the task. The comparators used for the experiments are justified and sufficient and the paper was well-written. The additional results during rebuttal helped clarify some of the concerns raised by the reviewers. While the new results will need to be included before the paper is published, there is strong interest in this line of work and the excitement on the proposed approaches is high.